# Computer code comprehension shares neural resources with formal logical inference in the fronto-parietal network

**Yun-Fei Liu\*, Judy Kim, Colin Wilson, Marina Bedny**

Johns Hopkins University, Baltimore, United States

**Abstract** Despite the importance of programming to modern society, the cognitive and neural bases of code comprehension are largely unknown. Programming languages might 'recycle' neurocognitive mechanisms originally developed for natural languages. Alternatively, comprehension of code could depend on fronto-parietal networks shared with other culturally-invented symbol systems, such as formal logic and symbolic math such as algebra. Expert programmers (average 11 years of programming experience) performed code comprehension and memory control tasks while undergoing fMRI. The same participants also performed formal logic, symbolic math, executive control, and language localizer tasks. A left-lateralized fronto-parietal network was recruited for code comprehension. Patterns of activity within this network distinguish between 'for' loops and 'if' conditional code functions. In terms of the underlying neural basis, code comprehension overlapped extensively with formal logic and to a lesser degree math. Overlap with executive processes and language was low, but laterality of language and code covaried across individuals. Cultural symbol systems, including code, depend on a distinctive fronto-parietal cortical network.

## Introduction

In 1800, only twelve percent of the world's population knew how to read, while today the world literacy rate is over eighty-five percent (https://ourworldindata.org/literacy). The ability to comprehend programming languages may follow a similar trajectory. Although only an estimated 0.5% of the world's population is currently proficient at computer programming, the number of jobs that require programming continues to grow. Coding is essential in scientific fields and in areas as diverse as artistic design, finance, and healthcare. As many industries incorporate artificial intelligence or other information technologies, more people seek to acquire programming literacy. However, the cognitive and neural mechanisms supporting coding remain largely unknown. Apart from its intrinsic and societal interest, programming is a case study of 'neural recycling' (*Dehaene and Cohen, 2007*). Computer programming is a very recent cultural invention that the human brain was not evolutionarily adapted to support. Studying the neural basis of code offers an opportunity to investigate how the brain performs novel complex skills.

Hypotheses about how the human brain accommodates programming range widely. One recently popular view is that code comprehension recycles mechanisms developed for human language (*Fedorenko et al., 2019*; *Fitch et al., 2005*; *Pandža, 2016*; *Portnoff, 2018*; *Prat et al., 2020*). Consistent with this idea, a recent study reported that individual differences in the ability to learn a second language predict aptitude in learning to program (*Prat et al., 2020*). Computer languages borrow letters and words from natural language and, in some programming languages like Python, the meanings of the borrowed symbols (e.g. `if`, `return`, `print`) relate to the meanings of the same symbols in English. As in natural languages, the symbols of code combine generatively according to a set of rules (i.e. a formal grammar). The grammars of language and that of code share

**\*For correspondence:**
yliu291@jhu.edu

**Competing interests:** The authors declare that no competing interests exist.

common features, including recursive structure (*Fitch et al., 2005*). In natural languages, a phrase can be imbedded within another phrase of the same syntactic category (*Hauser et al., 2002*; *Yang et al., 2017*). Analogously, in programming languages, data structures, such as lists and trees, can be recursive and a function can call itself. For example, in Python, IF conditionals can be embedded within IF conditionals:

```python
if (condition_1):
    if (condition_2):
        print('Both conditions are True.')
    else:
        print('Condition_1 is True, condition_2 is False.')
else:
    print('Condition_1 is False. Condition_2 not evaluated.')
```

To give another textbook example, the factorial of a positive integer N can be computed using a recursive Python function:

```python
def factorial(N):
    return N*factorial(N-1) if (N > 1) else 1
```

Here, the function `factorial` is called in the definition of itself. Given these similarities between programming languages and natural languages, one possibility then is that coding recycles neurocognitive mechanisms involved in producing and comprehending natural language. Other culturally-invented symbol systems, such as formal logic and mathematics do not appear to depend on the same neural network as natural language. Like code, formal logic and mathematics borrow symbols from language and are also hierarchical and recursive (e.g. (7*(7*(3+4)))). Unlike language, however, culturally-invented symbol systems are explicitly taught later in life. Computer coding, mathematics and logic, all involve manipulation of arbitrary variables without inherent meaning (e.g. `X, Y, input, ii`) according to a set of learned rules (*McCoy and Burton, 1988*). While each symbol system has its own conventionalized way of referring to variables and its own set of rules — indeed, these aspects differ somewhat among programming languages — there are nevertheless many common features. For example, conditional and other connectives (e.g. *'if…then'*, *'and'*, *'or'*, *'not'*) occur in both formal logic and programming languages with closely related meanings. Consider a function containing an `if` conditional written in Python,

```python
def fun(input):
    result = "result: "
    if input[0]=='a':
        result + = input[0].upper()
    return result
```

The value of the `result` variable depends on whether the `input` meets the specific conditions of the `if` statement. Similarly, in the logical statement '*If both X and Z then not Y*' the value of the result (*Y*) depends on the truth value of the condition '*both X and Z*'. One hypothesis, then, is that coding depends on similar neural resources as other culturally-invented symbol systems, such as formal logic and math.

Rather than recruiting perisylvian fronto-temporal areas, logic and math recruit a fronto-parietal network, including the dorsolateral prefrontal cortex (PFC) and the intraparietal sulcus (IPS) as well as putative symbol representations (i.e. numberform area) in inferior temporal cortex (*Amalric and Dehaene, 2016*; *Coetzee and Monti, 2018*; *Goel et al., 2007*; *Monti et al., 2009*). This fronto-parietal network overlaps partially with the so-called central executive/working memory system, which is implicated in a variety of cognitive tasks that involve maintaining and manipulating information in working memory, processes that are part and parcel of understanding and writing code (*Brooks, 1977*; *Duncan, 2010*; *Letovsky, 1987*; *Miller and Cohen, 2001*; *Soloway and Ehrlich, 1984*; *Weinberg, 1971*; *Zanto and Gazzaley, 2013*) (for a review of the cognitive models of code comprehension, see *Von Mayrhauser and Vans, 1995*). The central executive system is usually

studied using simple rule-based tasks, such as the multisource interference task (MSIT), Stroop task, and spatial or verbal working memory (*Banich et al., 2000*; *Bunge et al., 2000*; *Bush and Shin, 2006*; *January et al., 2009*; *Milham et al., 2001*; *Woolgar et al., 2011*; *Zanto and Gazzaley, 2013*; *Zhang et al., 2013*). Logic and math activate a similar network but also have unique neural signatures. Within the PFC, logic in particular recruits more anterior regions associated with more advanced forms of reasoning and symbol manipulation (*Coetzee and Monti, 2018*; *Ramnani and Owen, 2004*). The degree to which code comprehension relies on the same network as these other cultural symbol systems is not known.

Only a handful of previous studies have looked at the neural basis of code processing (*Duraes et al., 2016*; *Floyd et al., 2017*; *Ikutani and Uwano, 2014*; *Peitek et al., 2018*; *Siegmund et al., 2014*; *Huang et al., 2019*). Two studies observed larger fronto-parietal responses when comparing code writing and editing to prose writing and editing (*Floyd et al., 2017*; *Krueger et al., 2020*). When comprehension of code was compared to detection of syntactic errors in code, activity in both fronto-parietal and putative language areas was observed (*Siegmund et al., 2014*, *Siegmund, 2017*). None of these prior studies localized neural networks involved in language, or cultural symbol systems such as math and logic, in the same participants — leaving the question of direct neural overlap unanswered.

The goal of the current study was to ask whether basic computer code comprehension has a consistent neural signature across people, and, if so, whether this signature is more similar to those of other culturally-invented symbol systems (i.e. logic and math) or of natural language.

A group of expert programmers (average 11 years of programming experience) performed a code comprehension task while undergoing functional magnetic resonance imaging (fMRI). We chose a comprehension task partly because it could be analogous to understanding language vignettes and because it is arguably simpler than writing or debugging code. On each *real-code* trial, participants saw a short function definition, followed by an input and a possible output, and judged whether the output was valid. In *fake code* control trials, participants performed a memory task with unstructured text. A fake function was generated by scrambling a real function per line at the level of word/symbol. Each fake function preserved the perceptual and lexical elements of a real function, but was devoid of syntactic structure. The *real-code* condition contained two subtypes or 'control structures', for loops and if conditionals. We used multi-voxel-pattern analysis to decode for from if functions to test whether the code-responsive cortical system encodes code-relevant information. Finally, we examined the overlap of code comprehension with language (sentence comprehension), formal logic, and mathematical tasks. We also tested overlap of code with the MSIT to determine whether the overlap with culturally-invented symbol systems (i.e. logic and math) is more extensive than with simpler experimentally defined rule-based tasks.

## Results

### Behavioral results

Accuracy was similar across *real* and *fake code* trials (*real* M = 92%, SD = 0.045; *fake* M = 0.90, SD = 0.069; binary logistic mixed regression, *real* to *fake* odds ratio $\beta$ = 1.27; Wald's z statistic, z = 1.21; p=0.23). Accuracy was also similar across for and if trials (for M = 0.92, SD = 0.056; if M = 0.92, SD = 0.076; if to for odds ratio $\beta$ = 0.95; Wald's z statistic, z = −0.28; p=0.77). Participants were slower to respond to *fake* as compared to *real-code* trials (*real* M = 1.73 s, SD = 0.416; *fake* M = 2.03 s, SD = 0.37; t(73) = 2.329, p=0.023) and slower to respond to for as compared to if trials (for M = 1.85 s, SD = 0.46; if M = 1.60 s, SD = 0.44; t(58) = 2.127, p=0.038) (*Figure 1—figure supplement 1*).

In the language/math/logic localizer task, participants performed least accurately on logic trials, followed by math and language (logic M = 0.82, SD = 0.13; math M = 0.94, SD = 0.028; language M = 0.98, SD = 0.023; one-way-ANOVA, F(2, 42)=18.29, p<0.001). Participants were slowest to respond to logic trials, followed by math trials, and fastest on the language trials (logic M = 6.47 s, SD = 2.42; math M = 4.93 s, SD = 1.32; language M = 4.03, SD = 1.27; one-way-ANOVA F(2, 42) =7.42, p=0.0017) (*Figure 1—figure supplement 1*).

In the MSIT experiment, hard and easy conditions did not differ in terms of accuracy (hard M = 0.97, SD = 0.038; easy M = 0.98, SD = 0.034; t(28) = −1.363, p=0.18), but the hard trials took

significantly longer to respond to than the easy trials (hard M = 0.792 s, SD = 0.092; easy M = 0.506 s, SD = 0.090; t(28)=8.59, p<0.001) (*Figure 1—figure supplement 1*).

## fMRI results

## Code comprehension experiment

As compared to *fake code*, *real-code* elicited activation in a left-lateralized network of regions, including the lateral PFC (middle/inferior frontal gyri, inferior frontal sulcus; mainly BA 44 and 46, with partial activation in BA 6, 8, 9, 10, 47), the parietal cortex (the IPS, angular, and supramarginal gyri; BA 7) and the pMTG and superior temporal sulcus (BA 22 and 37). Activity was also observed in early visual cortices (Occ) (p<0.01 FWER, *Figure 1*; *Supplementary file 2*).

MVPA analysis revealed that for and if functions could be distinguished based on patterns of activity within PFC (accuracy = 64.7%, p<0.001), IPS (accuracy = 67.4%, p<0.001) and pMTG (accuracy = 68.4%, p<0.001). for and if functions could also be distinguished within the early visual cortex (accuracy = 55.7%, p=0.015), however, decoding accuracy was lower than in the other regions (F(3, 56)=4.78, p=0.0048) (*Figure 2*).

## Overlap between code comprehension and other cognitive domains

The language/math/logic localizer task activated previously identified networks involved in these respective domains. Responses to language were observed in a left perisylvian fronto-temporal language network, to math in parietal and anterior prefrontal areas as well as posterior aspect of the inferior temporal gyrus, and finally to logic, like math, in parietal and anterior prefrontal areas as well as posterior aspect of the inferior temporal gyrus. Logic activated more anterior and more extensive regions in PFC than math. The MSIT hard >easy contrast also activated a fronto-parietal network including the IPS, however, the activation in the lateral frontal cortex was posterior and close to the

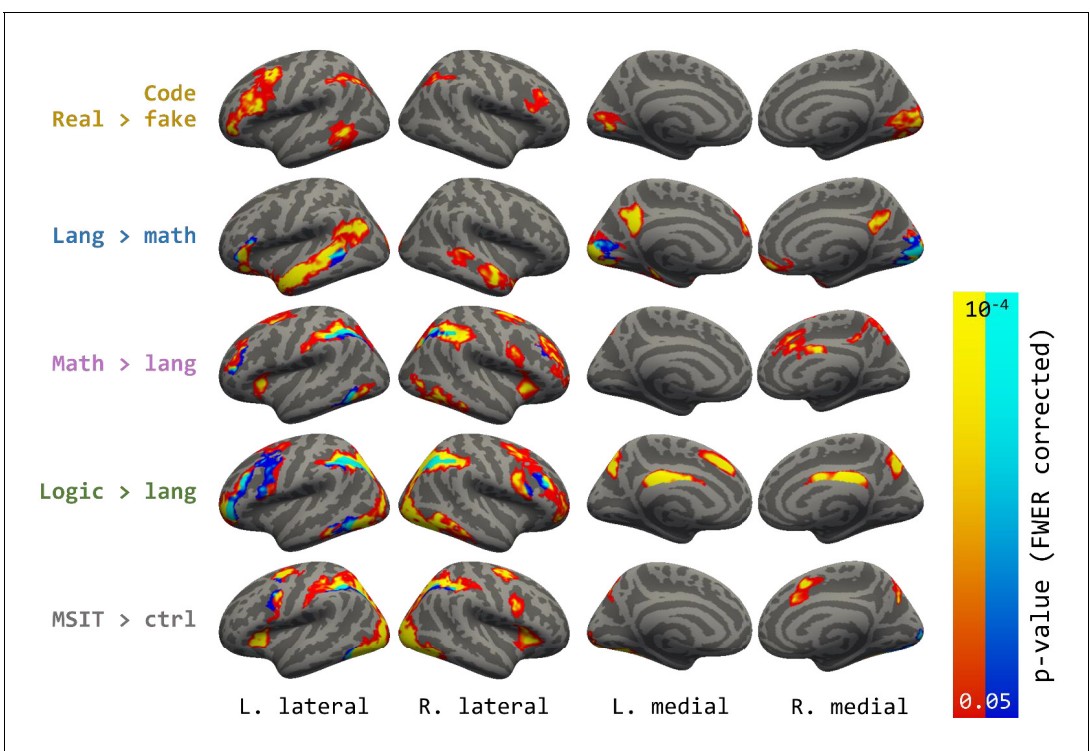

**Figure 1.** Whole-brain contrasts. Areas shown are p<0.05 cluster-corrected p-values, with intensity (both warm and cold colors) representing uncorrected vertex-wise probability. In the maps for each localizer contrast, both warm and cold colors indicate activated vertices in the contrast, with the cold color labelling the overlap with the code contrast.

The online version of this article includes the following figure supplement(s) for figure 1:

**Figure supplement 1.** Mean accuracy for (**a**) code comprehension and (**b**) localizer tasks.

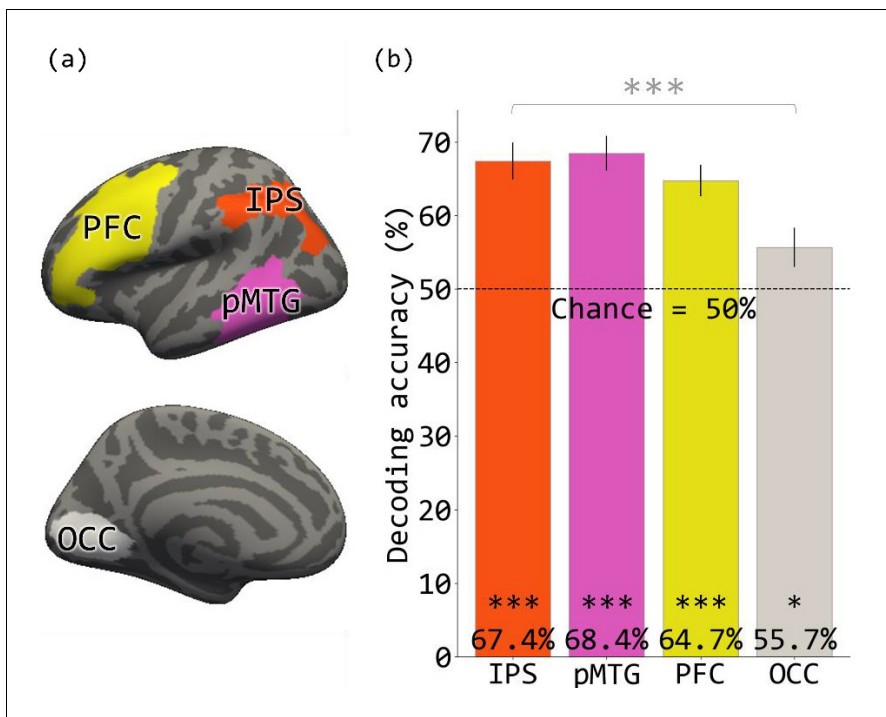

**Figure 2.** MVPA decoding accuracy in ROIs revealed by the code contrast. (**a**) The four search spaces (IPS, pMTG, PFC, OCC in the left hemisphere) within which functional ROIs were defined for the MVPA. (**b**) The MVPA decoding accuracy in the four ROIs. Error bars are mean ± SEM. *p<0.05. ***p<0.001.

precentral gyrus. (*Figure 3*, see *Supplementary file 2* for full description of activity patterns associated with language, logic, math and MSIT). Note that although in the current experiment logic, math and language were compared to each other, the networks observed for each domain are similar to those previously identified with other control conditions (e.g. lists of non-words for language and hard vs. easy contrast in a logic task) (e.g. *Coetzee and Monti, 2018*; *Fedorenko et al., 2011*).

Because code comprehension was highly left-lateralized, overlap analyses focused on the left hemisphere. Right hemisphere results are reported in the appendix. Code comprehension (*real >fake*) overlapped significantly above chance with all localizer tasks: logic, math, language and MSIT (each task compared to chance p's < 0.001, compared to code split-half overlap p's < 0.005) (*Figure 3*). The degree of overlap differed significantly across tasks (repeated-measures ANOVA: F (3,42) = 5.04, p=0.0045). Code comprehension overlapped most with logic (logic >language), followed by math and least with MSIT and language (*Figure 3*). Overlap with logic was significantly higher than with all other tasks, while the overlaps with the other three tasks (language, math, MSIT) were statistically indistinguishable from each other (post-hoc paired t-tests, FDR-corrected p's < 0.05) (*Supplementary file 3*). Note that overlap analyses control for the overall number of activated vertices in each contrast.

The overlap of code with logic and math was observed in the IPS, PFC, and a posterior portion of the inferior temporal gyrus (IT). PFC overlap was localized to the anterior middle frontal gyrus (aMFG, BA 46) and posteriorly in the precentral gyrus (BA 6). Overlap of code and the MSIT (hard >easy) was also observed in the IPS, precental gyrus and a small portion of the inferior temporal sulcus. Although MSIT and code overlapped in frontal and parietal areas, like code with logic/math, the precise regions of overlap within these general locations differed.

Finally, code overlapped with language (language >math) in portions of the inferior frontal gyrus and the posterior aspect of the superior temporal sulcus/middle temporal gyrus. The overlap between language and code was on average low, and the degree of overlap varied considerably across participants (cosine sim range: [0.105, 0.480]), with only half of the participants showing above chance overlap. Notably there was no relationship between overlap of code and language and level of expertise, as measured either by years of experience coding (regression against code-

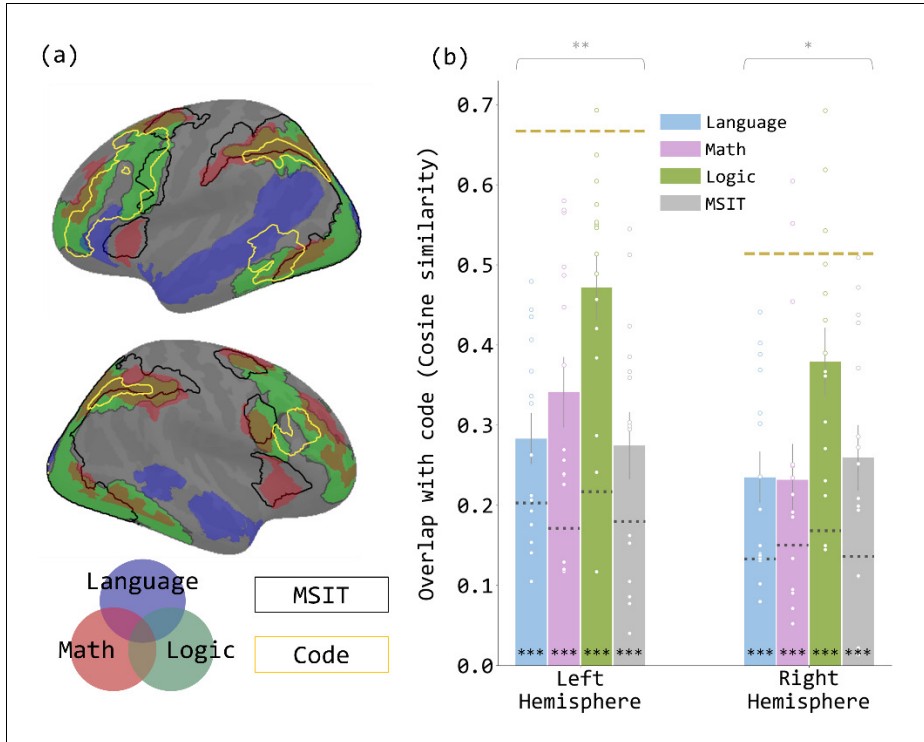

**Figure 3.** Overlap between the brain map revealed by the code contrast and each of the brain maps revealed by the localizer contrasts. (a) Brain map with the activated regions in the five contrasts reported in *Figure 1* overlain. The language network is shown in transparent blue, math in transparent red, and logic in transparent green. The regions activated in the MSIT contrast are enclosed in black outlines, and the code-responsive regions are enclosed in yellow outlines. (b) Cosine similarity between code contrast and each localizer contrast, in each hemisphere. Each dot represents the data from one participant. The dotted line on each bar indicates the null similarity between code contrast and the given localizer contrast. The yellow dashed line in each hemisphere indicates the empirical upper bound of the cosine similarity, the similarity between code comprehension and itself, averaged across participants. Error bars are mean ± SEM. *p<0.05. **p<0.01. ***p<0.001.

language overlap: $R^2 = 0$, p=0.99; regression against code-math overlap: $R^2 = 0.033$, p=0.52) or performance on coding assessments (regression against code-language overlap: $R^2 = 0.033$, p=0.52; regression against code-math overlap: $R^2 = 0.064$, p=0.36).

## Lateralization

The group activation map suggested that code comprehension is left-lateralized. Analyses of individual lateralization indices showed that indeed, code comprehension was as left-lateralized as language (Code lateralization index mean = 0.451, one sample t-test against 0: t(14) = 5.501, p<0.001; Language mean = 0.393, t(14) = 5.523, p<0.001; paired t-test between code and language: t(14) = 1.203, p=0.25). Moreover, lateralization indices of code and language were highly correlated across individuals ($R^2 = 0.658$, p<0.001) (*Figure 4*).

## Discussion

A consistent network of left-lateralized regions was activated across individuals during Python code comprehension. This network included the intraparietal sulcus (IPS), several regions within the lateral PFC and the posterior-inferior aspect of the middle temporal gyrus (pMTG). This code-responsive network was more active during *real* than *fake code* trials, even though for expert Python coders, the *fake code* task was more difficult (as measured by reaction time) than the *real-code* task. Involvement of the fronto-parietal system, as opposed to the fronto-temporal language network, in code

processing is consistent with prior work using more complex coding tasks, such as debugging and editing (*Siegmund et al., 2014*, *Siegmund, 2017*; *Huang et al., 2019*; *Krueger et al., 2020*; *Floyd et al., 2017*). The fact that fronto-parietal involvement is observed even for simple code function comprehension suggests that it is not related solely to cognitive control processes specific to these more complex coding tasks.

Within this code-responsive neural network, spatial patterns of activation distinguished between for vs. if code functions, suggesting that this network represents code-relevant information and is not merely activated during the coding task due to general difficulty demands. In overlap analyses, the code comprehension network was most similar to the fronto-parietal system involved in formal logical reasoning and to a lesser degree math. By contrast overlap with the perisylvian fronto-temporal language network is low. Notably, in the current study, neural responses associated with language, math and logic were localized partly relative to each other (e.g. logic >language). This approach focuses on networks that are uniquely involved in one domain vs. another. The degree to which areas shared across language, logic, and math are implicated in code comprehension remains to be addressed in future work.

## Code overlaps with logic

Code, logical reasoning, math and the MSIT task all activated aspects of the so-called fronto-parietal executive control system. However, overlap of code with logic was most extensive, followed by math and finally the MSIT. The difference between the MSIT task on the one hand and code comprehension, logic and math on the other, was particularly pronounced in the frontal lobe. There only code, logic and math activated more anterior regions of PFC, including BA 46 and BA 9, although logic-associated activation extended even more anteriorly than code. These findings suggest that neural overlap between logic and code is specific, and not fully accounted for by the general involvement of the central executive system. Note that although the logical reasoning task was more difficult than the language task, larger overlap with logic is unlikely to relate to task difficulty since the current overlap analyses control for the overall number of activated vertices.

Previous studies also find that the fronto-parietal network, including anterior prefrontal areas, are involved in logical reasoning (*Prado et al., 2011*; *Tsujii et al., 2011*). For example, anterior PFC is active when participants solve formal logical problems with quantifiers (e.g. 'all X are Y; Z is a X; therefore Z is Y') and connectives (e.g. 'if X then Y; not Y; therefore not X') and plays a key role in deductive reasoning with variables (*Coetzee and Monti, 2018*; *Goel, 2007*; *Goel and Dolan, 2004*; *Monti et al., 2009*; *Reverberi et al., 2010*; *Reverberi et al., 2007*; *Rodriguez-Moreno and Hirsch, 2009*).

A fronto-parietal network has also been consistently implicated in math (*Friedrich and Friederici, 2013*; *Maruyama et al., 2012*; *Piazza et al., 2007*; *Wendelken, 2014*). Some of the parietal responses to math have been linked to the processing of quantity information (*Eger et al., 2009*; *Nieder, 2016*; *Nieder and Miller, 2004*; *Piazza and Eger, 2016*; *Roitman et al., 2007*;

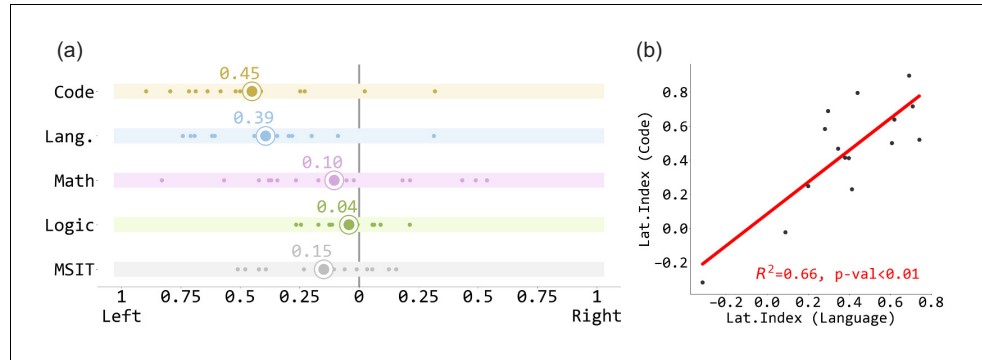

**Figure 4.** The lateralization index of the code contrast and the localizer contrasts. (a) The lateralization index of the code contrast and the localizer contrasts. Each white dot stands for one participant, and the enlarged dots represent the mean values. (b) The lateralization indices of code contrast and language contrast are highly correlated.

*Tudusciuc and Nieder, 2009*). For example, neurons in the IPS of monkeys, code numerosity of dots (*Nieder, 2016*). However, much of the same fronto-parietal network is also active during the processing of mathematical statements free of digits and arithmetic operations (*Amalric and Dehaene, 2018*; *Amalric and Dehaene, 2018*; *Wendelken, 2014*). In the current study, both the anterior prefrontal areas and parietal areas involved in math also overlapped with code and logical reasoning. Some of this activation could therefore reflect common operations, such as the manipulation of rules and symbols in working memory. On the other hand, the lower overlap between coding and math, as compared to coding and logic, could stem from math uniquely involving quantitative processing in the current study.

The present evidence suggests that culturally-invented symbol systems (i.e. code comprehension, formal logic and math) depend on a common fronto-parietal network, including the executive system. As noted in the introduction, although each of these symbol systems has its unique cognitive properties, they also have much in common. All involve the manipulation of abstract arbitrary symbols without inherent semantic content (e.g. `X, Y, input, result`) according to explicit rules. In the current logical inference and code experimental tasks, mental representations of several unknown variables are constructed (for logic '*X*', '*Y*', and '*Z*', for code '`input`' and '`result`') and the relationships between them deduced according to rules of formal logic or code.

There are also important differences between the rules of logical inference and programming. Take 'if' conditional judgement for example again. In formal logic, the statement '*if P then Q*' doesn't imply anything about what happens when *P* is false. On the contrary, in Python and most other programming languages, the statement.

```
if condition == True:
    do_something()
```

automatically implies that when the condition is false, the function 'do_something()" isn't executed, unless otherwise specified. Learning to program involves acquiring the particular set of conventionalized rules used within programming languages and a syntax that specifies how the programming language in question expresses logical operations (*Dalbey and Linn, 1985*; *Pea and Kurland, 1984*; *Pennington, 1987*; *Robins et al., 2003*). We speculate that such knowledge is encoded within the fronto-parietal network identified in the current study. It is also worth pointing out that although we found substantive overlap between the neural networks involved in code and logic, it is possible that, at a finer neural scale, these functions dissociate. Future studies comparing coders with different levels of expertise should test whether learning to code modifies circuits within the code-responsive neural network identified in the current study and address whether learning to code leads to specialization of a subset of this network for code in particular. A detailed understanding of the neural basis of code will also require development of cognitive models of code comprehension.

## The involvement of the multiple-demand executive control system in code comprehension

Code comprehension showed partial overlap with the MSIT task, particularly in the parietal cortex and in posterior frontal areas. Previous work has noted cognitive and neural similarity between arbitrary small-scale working memory tasks, such as the MSIT, and culturally-derived formal symbol systems (*Anderson, 2005*; *Qin et al., 2004*). As noted in the introduction, the MSIT task is a classic localizer for the executive function system (e.g. Stroop, n-back, and MSIT) (*Duncan, 2010*; *Fedorenko et al., 2013*; *Miller and Cohen, 2001*; *Woolgar et al., 2011*; *Zanto and Gazzaley, 2013*; *Zhang et al., 2013*). Like code comprehension, most experimental tasks that activate the central executive system involve the maintenance, manipulation and selection of arbitrary stimulus response mappings according to a set of predetermined rules (*Woolgar et al., 2011*; *Zhang et al., 2013*). For example, in the MSIT task among the many possible ways to map a visually presented digit triplet to a button press, participants maintain and select the rule 'press the button whose index corresponds to the value of the unique digit in the triplet.' The difficult condition requires using this less habitual rule to make a response.

Previous studies also find that the fronto-parietal executive system is involved in rule maintenance and switching, as well as variable representation. In one task-switching study, participants

maintained a cued rule in working memory and the level of fronto-parietal activity increased with the complexity of the rule maintained (*Bunge et al., 2003*). Patterns of neural activity within the executive system encoded which rule is currently being applied and activity is modulated by rule switching (*Buschman et al., 2012*; *Crittenden and Duncan, 2014*; *Xu et al., 2017*). Finally, studies with non-human primates find that neurons in the frontal lobe encode task-based variables (*Duncan, 2010*; *Kennerley et al., 2009*; *Nieder, 2013*). Such processes, studied in the context of simple experimental tasks, may also play a role in code comprehension.

Although culturally-invented formal symbol systems and simple experimental rule-based tasks (e.g. MSIT) share cognitive elements, the latter, unlike the former, involve simple rules that specify stimulus response mappings, rather than mental manipulations of variables. An intriguing possibility is that code comprehension and other culturally-invented symbol systems recycle a subset of the neural machinery that originally evolved for the maintenance and manipulation of simpler variables and rules (*Anderson, 2005*; *Qin et al., 2004*).

## Code comprehension and language

In the current study, perisylvian fronto-temporal network that is selectively responsive to language had low and variable overlap with the neural network involved in code comprehension. The regions that did show some overlap between language and code (i.e. left inferior frontal and middle temporal gyri), have been implicated in high-level linguistic processing, including sentence-level syntax (*Friederici, 2017*; *Hagoort, 2005*; *Pallier et al., 2011*; *Bornkessel-Schlesewsky and Schlesewsky, 2013*; *Fedorenko and Thompson-Schill, 2014*; *Matchin and Hickok, 2020*). The current results therefore do not rule out the possibility that the language system plays some role in code. Nevertheless, on the whole, the results do not support the hypothesis that the language system is recycled for code comprehension (see also Ivanova et al, in press). Previous studies also find that math and formal logic do not depend on classic language networks (*Amalric and Dehaene, 2016*; *Monti et al., 2009*). The low degree of overlap between code and language is intriguing given the cognitive similarities between these domains (*Fedorenko et al., 2019*; *Pandža, 2016*; *Peitek et al., 2018*; *Portnoff, 2018*; *Prat et al., 2020*; *Siegmund et al., 2014*). As noted in the introduction, programming languages borrow letters and words from natural language, and both natural language and code have hierarchical, recursive grammars (*Fitch et al., 2005*).

One possible explanation for low overlap between the perisylvian fronto-temporal language network and code, is that the language system is evolutionarily predisposed to support natural language processing in particular, and is therefore not generalizable even to similar domains, like computer code and formal logic (*Dehaene-Lambertz et al., 2006*; *Fedorenko et al., 2011*). Timing could also play a role. The perisylvian fronto-temporal language network may have a sensitive period of development during which it is most capable of learning (*Cheng et al., 2019*; *Mayberry et al., 2018*; *Cheng et al., 2020*; *Ferjan Ramirez et al., 2016*) By the time people learn to code, the network may be incapable of taking on new cognitive functions. Indeed, even acquiring a second language late in life leads to lower levels of proficiency and responses outside the perisylvian fronto-temporal system (*Hartshorne et al., 2018*; *Johnson and Newport, 1989*; ). These observations suggest that domain-specific systems, like the perisylvian fronto-temporal language network, are not always amenable for 'recycling' by cultural inventions. The fronto-parietal system might be inherently more flexible throughout the lifespan and thus more capable of taking on new cultural skills (*Riley et al., 2018*).

Despite lack of direct overlap, lateralization patterns of language and coding were highly correlated across individuals that is those individuals with highly left-lateralized responses to sentences also showed highly left-lateralized responses to code. This intriguing observation suggests that the relationship between code and language may be ontogenetic as well as phylogenetic. It is hard to imagine how code in its current form could have been invented in the absence of language (*Fitch et al., 2005*). Ontogenetically, code-relevant neural representations might be enabled by the language system, even though they are distinct from it.

An analogous example comes from the domain of reading (*Dehaene et al., 2010*; *McCandliss et al., 2003*). Reading-relevant regions, such as the visual word form area (VWFA), are co-lateralized with the perisylvian fronto-temporal language network across people (*Cai et al., 2010*). The VWFA has strong anatomical connectivity with the fronto-temporal language network even prior to literacy (*Bouhali et al., 2014*; *Saygin et al., 2016*). Analogously, code comprehension

may colonize a left-lateralized portion of the central executive system due to its stronger (i.e. within hemisphere) connectivity with the perisylvian fronto-temporal language network.

## Relationship to co-published work by Ivanova and colleagues

The current results are consistent with the co-published study by Ivanova and colleagues. Like the current study, Ivanova et al report the involvement of a fronto-parietal network in code comprehension and low overlap with fronto-temporal language processing systems. The consistency of these two studies is striking in light of the difference in the study designs. While the current study compared code comprehension to a working memory control task with linguistic symbols, Ivanova et al. compared code comprehension to matched verbal descriptions of program-like algorithms. In the current study, the value of input variables was provided only after function presentation, by contrast, the value was stipulated as part of the function in Ivanova et al. While the current study localized the language network using a passive/active sentence comparison task relative to a math control condition, Ivanova and colleagues compared sentence comprehension to a non-word memory control task. Finally, in addition to examining the neural basis of Python comprehension, Ivanova et al also studied the neural basis of ScratchJr and found partially similar results.

There are also interesting differences across experiments that may relate to differences in design. Both the current study and Ivanova et al's Python comprehension tasks revealed robust responses on prefrontal cortices, by contrast prefrontal responses to ScratchJr were weaker. Unlike Python, ScratchJr doesn't declare variables and update their values. Previous studies have implicated anterior prefrontal cortices in variable manipulation (*Monti et al., 2009*; *Diester and Nieder, 2007*). The degree of prefrontal involvement in code comprehension may therefore depend in part on whether variable manipulation is involved. In the current study, we observed strong left-lateralization of code comprehension and co-lateralization of code comprehension and language across people. By contrast, Ivanova and colleagues did not observe left-lateralization of code comprehension. Since Ivanova and colleagues compared code comprehension to a sentence reading task, left-lateralization may have been obscured by subtracting out a left-lateralized pattern associated with sentence processing. Alternatively, the Ivanova study focused on the so- called 'syntactic' aspects of code comprehension rather than the semantics of code by comparing code comprehension to matched sentences describing similar algorithms. It is possible that the semantics or algorithmic aspects of code are more left-lateralized. Notably, since the current sample is relatively small (n = 15), any differences between the current study and the Ivanova findings should be interpreted with caution and replicated in future work.

## Conclusions

A fronto-parietal cortical network is consistently engaged in expert programmers during code comprehension. Patterns of activity within this network distinguish between FOR and IF functions. This network overlaps with other culturally-invented symbol systems, in particular formal logic and to a lesser degree math. By contrast, the neural basis of code is distinct from the perisylvian fronto-temporal language network. Rather than recycling domain-specific cortical mechanisms for language, code, like formal logic and math, depends on a subset of the domain general executive system, including anterior prefrontal areas. The executive system may be uniquely suited as a flexible learning mechanism capable of supporting an array of cultural symbol systems acquired in adulthood.

## Materials and methods

### Participants

Seventeen individuals participated in the study; one did not complete the tasks due to claustrophobia, and another was excluded from analyses due to excessive movement (>2 mm). We report data from the remaining fifteen individuals (three women, age range 20–38, mean age = 27.4, SD = 5.0). All participants had normal or corrected to normal vision and had no known cognitive or neurological disabilities. Participants gave informed consent according to procedures approved by the Johns Hopkins Medicine Institutional Review Board (IRB protocol number: NA_00087983).

All participants had at least 5 years of programming experience (range: 5–22, mean = 10.7, SD = 5.2), and at least 3 years of experience with Python (range: 3–9, mean = 5.7, SD = 1.8).

## Behavioral pre-test

In addition to self-reported programming experience, Python expertise was evaluated with two out-of-scanner Python exercises (one easier and one more difficult) the week prior to the fMRI experiment. These exercises also served to familiarize participants with the particular Python expressions that would be used during the fMRI experiment.

The easier exercise consisted of three phases. During the first phase (*initial test*), we evaluated participants' knowledge of every built-in Python function that would appear in the stimuli of the fMRI experiment. Participants were asked to type the output of a single-line `print()` statement (e.g. for 'print("3.14'.split('1'))' one should type '['3.', '4']'). On average participants answered M = 82.9% (SD = 6.9%) of the questions correctly (range: 70–96%). Since even expert programmers may not have used a particular function in the recent past, the second phase (*recap)* explicitly reviewed the definitions and purposes of all of the relevant functions and expressions. During the final phase (*retest*), participants were once again asked to type the output of a single-line statement for each function (M = 92.0% (SD = 7.5%), range: 72.4–100%).

The more difficult exercise evaluated the participants' knowledge about when and how to use Python functions and expressions. Each participant answered sixteen questions consisting of a code snippet with a blank. A prompt was presented alongside the code snippet to explain what the snippet should output if executed. The participant was asked to fill in the blank in order to complete the code (see the subsection 'example of the difficult out-of-scanner exercise' in the appendix). The questions were designed by the experimenter to cover some of the objectives specified in the exam syllabus of the Certified Associate in Python Programming Certification held by the Python Institute (https://pythoninstitute.org/certification/pcap-certification-associate/pcap-exam-syllabus/). On average, the participants got 64.6% (SD = 16.6%) of the questions correct (range: 37.5–93.75%).

## fMRI task design and stimuli

### Code comprehension experiment

In *real-code* comprehension trials, participants were presented with Python functions designed for the purposes of the experiment. In *fake code* control trials, they were presented with incomprehensible scrambled versions of the same functions (for details on real and fake code, see below). To help participants distinguish between *real* and *fake code* trials and to prevent the participants from erroneously attempting to parse fake code, *real-code* appeared in white text and *fake code* in yellow text.

Each trial had three phases: *function* (24 s), *input* (6 s), and *question* (6 s) (**Figure 5**). First, participants viewed a Python function for 24 s, followed by a 500 millisecond fixation-cross delay. During the *input* phase, the original code function re-appeared on the screen with a potential input below consisting of a single-line character string (6 s). Participants were instructed to use the input to mentally derive the output of the function shown during the *input* phase. After the *input* phase there was a 500 millisecond fixation-cross delay followed by a proposed output along with the prompt '*TRUE?*' Participants were asked to determine whether the output was correct within 6 s. All trial phases had a shortening bar at the bottom of the screen indicating the remaining time during that particular phase of the trial. Each trial was followed by a 5 s inter-trial interval during which the text '*Your response is recorded. Please wait for the next trial.*' was shown on the screen.

Each *real-code* function consisted of five lines. The first line (`def fun(input):`) and the last (`return result`) were always the same. The second line always initialized the result variable, and the third and fourth lines formed a control structure (either a `for` loop or an `if` conditional) that may modify the value of the result. *real-code* trials were divided into two sub-conditions, `for` and `if`, according to the control structures the functions contained. Each condition included two variants of the for or if functions (see the subsections 'detailed information about the stimuli' and 'the two variants of each control structure' in the appendix). All functions took a letter string as input and performed string manipulation.

*Fake code* trials were analogous to the *real-code* trials in temporal structure (i.e. function, input, question). However, no *real-code* was presented. Instead, participants viewed scrambled text and were asked to remember it. During the *function* phase of a *fake code* trial, participants saw a scrambled version of a real-code function. Scrambling was done within line at word and symbol level (**Figure 5**, bottom row). Because fake functions were derived from real functions, the words, digits and

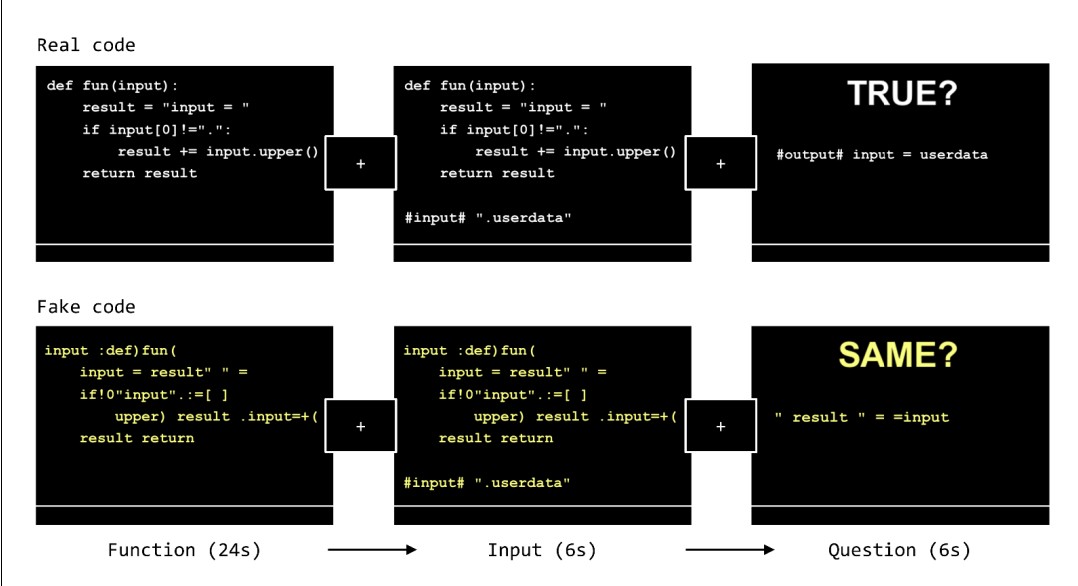

**Figure 5.** The experiment design. The FAKE function (bottom row) in this figure is created by scrambling the words and symbols in each line of the REAL function (top row). Note that for the purpose of illustration, the relative font size of the text in each screen shown in this figure is larger than what the participants saw during the actual MRI scan.

operators that existed in real functions were preserved; however, none of the scrambled lines comprised an executable Python statement. During the *input* phase, an additional fake input line appeared below the fake function. The fake input line didn't interact with the fake function, the participants only had to memorize this line. During the *question* phase, a new character line appeared along with the prompt '*SAME?*' Participants judged whether this line had been presented during the *function* and *input* phases (including the additional input line), or it came from a different fake function. The correct response was 'true' for half of the *real-code* trials and half of the *fake code* trials.

There were six task runs, each consisting of 20 trials, eight *real if code*, eight *real for code* and four *fake* code trials. Each participant saw a total of 48 *for* functions (24 per variant), 48 *if* functions (24 per variant), and 24 *fake* functions (12 fake for, and 12 fake if functions). After each run of the task, participants saw their overall percent correct and average response time. Participants were divided into two groups such that the variants of the functions were counterbalanced across groups; the same participant never saw different variants of the same function. The order of the presentation of the functions was pseudo-randomized and balanced across participants. In total, 192 *real* functions (96 per group) and 48 *fake* functions (24 per group) were used in the experiment. All the functions are listed in *Supplementary file 1*. We permuted the order of the functions systematically such that each participant saw a unique order (see the subsection 'algorithm for stimulus permutation' in the appendix).

## Localizer Tasks

During a separate MRI session, participants took part in two localizer experiments. A single experiment was used to localize responses to formal logic, symbolic math, and language using each condition as the control for the others: logic/math/language localizer. The task design was adapted from *Monti et al., 2009*, *Monti et al., 2012* (*Kanjlia et al., 2016*; *Monti et al., 2009*; *Monti et al., 2012*). On language trials, participant judged whether two visually presented sentences, one in active and one in passive voice, had the same meaning (e.g. '*The child that the babysitter chased ate the apple*' vs '*The apple was eaten by the babysitter that the child chased*'). On math trials, participant judged whether the variable X had the same value across two equations (e.g. '*X minus twenty-five equals forty-one*' vs '*X minus fifty-four equals twelve*'). On formal logic trials, participant judged whether two logical statements were consistent, where one statement being true implied the other also being true (e.g. '*If either not Z or not Y then X*' vs '*If not X then both Z and Y*').

Each trial began with a 1 s fixation cross. One member of a pair appeared first, the other following 3 s later. Both statements remained on the screen for 16 s. Participants pressed the right or left button to indicate true/false. The experiment consisted of 6 runs, each containing 8 trials of each type (language/math/logic) and six rest periods, lasting 5 s each. All 48 statement pairs from each condition were unique and appeared once throughout the experiment. In half of the trials, the correct answer was 'true'. Order of trials was counterbalanced across participants in two lists.

Although all of the tasks in the language/math/logic localizer contain language stimuli, previous studies have shown that sentences with content words lead to larger responses in the perisylvian fronto-temporal language network than spoken equations or logical statements with variables (*Kanjlia et al., 2016*; *Monti et al., 2009*; *Monti et al., 2012*). The perisylvian fronto-temporal language network shows enhanced activity for stimuli that contain meaningful lexical items *and* sentence-level syntax (e.g. *Fedorenko et al., 2016*). Furthermore, previous studies have found that responses to language, logic and math when compared to each other were similar to what was observed for each domain relative to independent control conditions (e.g. sentences relative to lists of non-words for language, and easy vs. hard logic problems; *Kanjlia et al., 2016*; *Monti et al., 2009*, *Monti et al., 2012*).

The multi-source interference task (MSIT) was adapted from *Bush and Shin, 2006* to engage executive control processes and localize the multiple-demand network. On each trial, a triplet of digits was shown on the screen, two of which were the same. The participant pressed a button (1, 2, or 3) to indicate the identity of the target digit that was different from the distractors. For example, for '131' the correct response is '3'; for '233' it is '2'. The participants always pressed buttons '1', '2', and '3' with their index, middle, and ring fingers, respectively.

MSIT consisted of *interference* blocks and *control* blocks, each containing 24 trials (1.75 s each). On interference trials, the location of the target digit was inconsistent with the identity of the digit (e.g. trials such as '133' or '121' did not occur). On control trials, the distractors were always '0', and the target digit was always at the same location as its identity. In other words, there were only three kinds of control trial, namely '100', '020', and '003'.

Participants performed 2 runs of MSIT. Each run began with 15 s of fixation, followed by four interference blocks and four control blocks interleaved, and ended with another 15 s of fixation. Each block lasted 42 s. The order of the blocks was balanced within and between participants. Trial order was arranged such that all 12 interference trials appeared exactly twice in an interference block, and all three control trials appeared exactly six times in a control block. Identical trials never appeared in succession, and the order of the trials was different across all 8 blocks of the same kind.

## Data acquisition

MRI data were acquired at the F.M. Kirby Research Center of Functional Brain Imaging on a 3T Phillips Achieva Multix X-Series scanner. T1-weighted structural images were collected in 150 axial slices with 1 mm isotropic voxels using the magnetization-prepared rapid gradient-echo (MP-RAGE) sequence. T2*-weighted functional BOLD scans were collected in 36 axial slices (2.4 $\times$ 2.4$\times$3 mm voxels, TR = 2 s). We acquired the data in one code comprehension session (six runs) and one localizer session (2 runs of MSIT followed by 6 runs of language/math/logic), with the acquisition parameters being identical for both sessions.

The stimuli in both the code comprehension and localizer sessions were presented with custom scripts written in PsychoPy3 (https://www.psychopy.org/, *Peirce et al., 2019*). The stimuli were presented visually on a rear projection screen, cut to fit the scanner bore, with an Epson PowerLite 7350 projector. The resolution of the projected image was 1600 $\times$ 1200. The participant viewed the screen via a front-silvered, 45°inclined mirror attached to the top of the head coil.

## fMRI data preprocessing and general linear model (GLM) analysis

Data were analyzed using Freesurfer, FSL, HCP workbench, and custom in-house software written in Python (*Dale et al., 1999*; *Smith et al., 2004*; *WU-Minn HCP Consortium et al., 2013*). Functional data were motion corrected, high-pass filtered (128 s), mapped to the cortical surface using Freesurfer, spatially smoothed on the surface (6 mm FWHM Gaussian kernel), and prewhitened to remove temporal autocorrelation. Covariates of no interest were included to account for confounds related to white matter, cerebral spinal fluid, and motion spikes.

The four *real-code* (`for1`, `for2`, `if1`, `if2`) and corresponding *fake code* conditions were entered as separate predictors in a GLM after convolving with a canonical hemodynamic response function and its first temporal derivative. Only the images acquired during the twenty-four-second *function* phase were modeled.

For the localizer experiment, a separate predictor was included for each of the three conditions (language, math, and logic) modeling the 16 s during which the statement pair was presented, as well as a rest period (5 s) predictor. In the MSIT task, the interference condition and the control condition were entered as separate predictors.

Each run was modeled separately, and runs were combined within each subject using a fixed-effects model (*Dale et al., 1999*; *Smith et al., 2004*). For the group-level analysis across participants, random-effects models were applied, and the models were corrected for multiple comparisons at vertex level with p<0.05 false discovery rate (FDR) across the whole brain. A nonparametric permutation test was further implemented to cluster-correct at p<0.01 family-wise error rate.

## ROI definition

For each participant, four code-responsive functional ROIs were defined to be used in the MVPA analysis. First, random-effects whole-brain univariate analysis for the *real >fake code* contrast revealed four major clusters in the left hemisphere: the intraparietal sulcus (IPS), the posterior middle temporal gyrus (pMTG), the lateral PFC, and the early visual cortex (Occ). These clusters were used to define group search spaces. Each search space was defined by combining parcels from Schaefer et al. that encompassed each cluster (400-parcel map, *Schaefer et al., 2018*). Next, individual functional ROIs were defined within these clusters by taking the top 500 active vertices for the *real >fake* contrast within each participant.

## MVPA

MVPA was used to distinguish for and if functions based on the spatial activation pattern in code-responsive ROIs. Specifically, we used the support vector machine (SVM) implemented in the Python toolbox Scikit-learn (*Chang and Lin, 2011*; *Pedregosa et al., 2011*).

For each participant, the spatial activation pattern for each function was defined as the beta parameter estimation of a GLM with each function entered as a separate predictor. Within each ROI in each participant, the 96 spatial patterns elicited by the real functions were collected. Normalization was carried out separately for the for condition and if condition such that in either condition, across all vertices and all functions, the mean was set to 0 and standard deviation to 1. The purpose of the normalization is to eliminate any difference in the baselines of the two conditions while preserving distinctive spatial patterns.

The whole dataset was split into a training test (90%, 86 functions) and a testing set (10%, 10 functions), where in each set, half of the patterns came from for functions. A linear SVM (regularization parameter C = 5.0) was trained on the training set and tested on the testing set. Classification was carried out on 100 different train-test splits, and the average accuracy value was recorded as the observed accuracy.

We tested the classifier performance against chance (50%) using a combined permutation and bootstrapping approach (*Schreiber and Krekelberg, 2013*; *Stelzer et al., 2013*). We derived the t-statistic of the Fisher-z transformed accuracy values against chance (also Fisher-z transformed). The null distribution for each participant was generated by first shuffling the condition labels 1000 times, then computing the mean accuracy derived from the 100 train-test split of each shuffled dataset. Then, a bootstrapping method was used to generate an empirical distribution of the t-statistics. In each of the $10^6$ iterations of the bootstrapping phase, one Fisher-z transformed null accuracy value (out of 1,000) per participant was randomly selected, and a one sample t-test was applied to the null sample. The empirical p-value of the real t-statistic was defined as the proportion of the null t-statistics greater than the real value.

## Overlap analysis

For each participant, and in each hemisphere, we used cosine similarity to quantify the overlap of the activated vertices between code comprehension and each of the four localizer contrasts: language (language >math), math (math >language), logic (logic >language), and multi-source

interference (hard >easy). First, we generated the binary activation map for each contrast. A vertex was assigned the value one if the significance of its activation is above the 0.05 (FDR-corrected) threshold, and 0 otherwise. Each binary map was regarded as a vector, and the cosine similarity between two vectors (e.g. code comprehension and logic) was defined as the inner product of the vectors divided by the product of their respective lengths (norms). Note that this measure controls for overall vector length (i.e. the overall number of active voxels in each contrast). The cosine similarities of code to each of the localizer tasks was then compared using repeated-measure ANOVA and post-hoc pairwise comparisons with false discovery rate (FDR) correction.

The empirical lower bound was calculated separately for each localizer task to account for differences in the number of activated vertices across tasks. For each participant, for each localizer task, we computed the cosine similarity between the binary map for code comprehension and a shuffled binary map for each localizer task. This step was repeated 100 times to generate the null distribution of the similarity values.

We used a bootstrapping approach to test whether each observed cosine similarity value was significantly above the empirical lower bound. For each localizer task, we randomly selected one similarity value from the null distribution of one participant and computed a null group mean similarity. This step was repeated $10^6$ times to derive the null distribution of the null group mean similarity. The empirical p-value of the real group mean similarity was defined as the proportion of the null values greater than the real value.

We operationalized the empirical upper bound as the cosine similarity of code comprehension and itself. For each participant, we split the data for code comprehension in half, ran a GLM for each half, and derived two binary maps whose cosine similarity was computed. We averaged all the similarity values resulting from the 10 possible splits of the six runs and across all participants.

## Additional information

### Funding

| Funder | Author |
| --- | --- |
| Johns Hopkins University | Marina Bedny |

The funders had no role in study design, data collection and interpretation, or the decision to submit the work for publication.

### Author contributions

Yun-Fei Liu, Conceptualization, Data curation, Software, Formal analysis, Investigation, Visualization, Methodology, Writing - original draft, Project administration, Writing - review and editing; Judy Kim, Conceptualization, Resources, Methodology, Writing - review and editing; Colin Wilson, Conceptualization, Resources, Software, Formal analysis, Methodology, Writing - review and editing; Marina Bedny, Conceptualization, Resources, Supervision, Funding acquisition, Validation, Methodology, Project administration, Writing - review and editing

### Author ORCIDs

Yun-Fei Liu (ID) https://orcid.org/0000-0001-6644-813X

### Ethics

Human subjects: Participants gave informed consent according to procedures approved by the Johns Hopkins University Institutional Review Board. (IRB protocol number: NA_00087983).

### Decision letter and Author response

Decision letter https://doi.org/10.7554/eLife.59340.sa1
Author response https://doi.org/10.7554/eLife.59340.sa2

## Additional files

### Supplementary files

- Supplementary file 1. All the functions included in the fMRI study.
- Supplementary file 2. Activated clusters in each contrast.
- Supplementary file 3. FDR-corrected p-values for the post-hoc paired t-tests among the overlap between code contrast and the localizer contrasts.
- Transparent reporting form

### Data availability

Data includes human brain imaging; therefore, they can't be posted online without prior consents from the participants. De-identified behavioral data has been posted on OSF.

The following dataset was generated:

| Author(s) | Year | Dataset title | Dataset URL | Database and Identifier |
|---|---|---|---|---|
| Liu Y | 2020 | The neural basis of code comprehension | https://osf.io/58mwu | Open Science Framework, 10.17605/OSF.IO/58MWU |

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

## Appendix

An example of the difficult out-of-the-scanner exercise

```
Consider the following code:
###

def xx(a):
    b = a+3
    return
if xx(1)==________:
    print("HEY!!")

###
```

```
What keyword should be filled in the blank if we want the code to print
"HEY!!"?
(It's a keyword. There should be only letters in your answer.)
```

### Detailed information about the stimuli

Across all real functions, the variable names input, result, and `ii`, were the same, and only the 12 built-in functions (`capitalize()`, `isalnum()`, `isalpha()`, `isdigit()`, `len()`, `lower()`, `range()`, `sorted()`, `split()`, `str()`, `swapcase()`, and `upper()`) and three expressions (list comprehension, slice notation, and string formatting, see *Supplementary file 1* for examples) tested during the screening exercise were included in the user-defined functions.

The addition operator (+) occurred in all functions, but always meant string concatenation rather than numeric addition, and never took a numeric as operand. In each group, the multiplication operator (*) existed in 32 out of the 96 real functions, and 10 of them took a numeric as one of its operands. However, in all these instances, the 'multiplication' meant repetition of strings or lists instead of numeric multiplications (e.g. `'abc'*3` results in `abcabcabc`). In each group, 12 out of the 96 real functions contained a comparison to a numeric value, such as '`len(input)>5`'.

### The two variants of each control structure

We designed two variants to implement each control structure, `for` and `if`. In the first variant of a `for` code, the `for` loop was implemented in the canonical way. In the second variant of a `for` code, we implemented the loop with a Python-specific expression 'list comprehension', where the operation to be performed on each element in a list (or string) was stated before specifying the list to be iterated over. In the first variant of an `if` code, the if conditional was implemented in the canonical way. In the second variant of an `if` code, the conditional was implemented by first stating the action to take `if` a condition is true, then multiplying this action to the true/false judgement statement of the condition. There was not a formal jargon for this kind of implementation, for the sake of convenience, we called it 'conditional multiplication' in this study. Please refer to *Supplementary file 1* for examples of each variant.

### The algorithm for stimulus permutation

In this experiment, there were five conditions, 'FOR1', 'FOR2', 'IF1', 'IF2', and 'FAKE'. For simplicity, from here on we label them as 'A', 'B', 'C', 'D', and 'E', respectively.

There were 120 permutations for five distinct labels, such as 'ABCDE', 'BCDEA', 'CDEAB', 'DEABC', 'EABCD', 'ACBDE', 'CBDEA', etc. Each run consisted of 20 functions, which was 4 permutations of 5 labels. Therefore, for each run, we drew four permutations out of the 120 possible permutations. So, the order a participant saw in the first run can be:

ABCDE BCDEA CDEAB DEABC

And the order in the second run can be:

EABCD ACBDE CBDEA BDEAC

The permutations were allocated such that every participant saw 24 permutations across all six runs, and every five participants saw all the 120 permutations.

After determining the order of the conditions, we assigned actual instances of the conditions to the labels. The order of presentation for the functions in each condition was also permuted such that a function in run 1 for participant one appeared in run 2 for participant 2, run 6 for participant 6, run 1 for participant 7, and so on. Specifically, the first run of the first participant could be:

$A_1 B_1 C_1 D_1 E_1 \ B_2 C_2 D_2 E_2 A_2 \ C_3 D_3 E_3 A_3 B_3 \ D_4 E_4 A_4 B_4 C_4$

While the first run of the second participant could be:

$A_5 B_5 C_5 D_5 E_5 \ B_6 C_6 D_6 E_6 A_6 \ C_7 D_7 E_7 A_7 B_7 \ D_8 E_8 A_8 B_8 C_8$

The second participant still saw $A_1$, $B_1$, $C_1$, . . . . . . $D_4$, $E_4$, just in some later runs.

As a result of permutations of both conditions and functions within condition, all of the participants saw a unique order of presentation.

## Overlap analysis in the right hemisphere

Code comprehension (*real >fake*) overlapped significantly above chance with all localizer tasks: logic, math, language and MSIT (each task compared to chance $p$'s $< 10^{-6}$ compared to code split-half overlap $p$'s $< 10^{-6}$). The degree of overlap differed significantly across tasks (repeated-measures ANOVA: $F(3,42) = 3.03$, $p=0.040$). Post-hoc paired t-tests (FDR-corrected $p$'s $< 0.05$) revealed that the overlap with logic was significantly higher than with math and MSIT, but indistinguishable from the overlap with language. The overlaps with the other there tasks (language, math, MSIT) were statistically indistinguishable from each other (*Supplementary file 3*).

