## [Decision Letter]

**Decision letter after peer review:**

Thank you for submitting your article "Computer code comprehension shares neural resources with formal logical inference in the fronto-parietal network" for consideration by *eLife*. Your article has been reviewed by three peer reviewers, and the evaluation has been overseen by a Reviewing Editor and Timothy Behrens as the Senior Editor. The following individuals involved in review of your submission have agreed to reveal their identity: William Matchin (Reviewer #1); Ina Bornkessel-Schlesewsky (Reviewer #3).

The reviewers have discussed the reviews with one another and the Reviewing Editor has drafted this decision to help you prepare a revised submission.

First, thank you for taking part in the review process.

As you know, *eLife* is invested in changing scientific publishing and experimenting to embody that change, even if that involves a degree of risk in order to find workable changes. In this spirit, the remit of the co-submission format is to ask if the scientific community is enriched by the data presented in the co-submitted manuscripts together more so than it would be by the papers apart, or if only one paper was presented to the community. In other words, are the conclusions that can be made are stronger or clearer when the manuscripts are considered together rather than separately? We felt that despite significant concerns with each paper individually, especially regarding the theoretical structures in which the experimental results could be interpreted, that this was the case.

We want to be very clear that in a non-co-submission case we would have substantial and serious concerns about the interpretability and robustness of the Liu et al. submission given its small sample size. Furthermore, the reviewers' concerns about the suitability of the control task differed substantially between the manuscripts. We share these concerns. However, despite these differences in control task and sample size, the Liu et al., and Ivanova et al. submissions nonetheless replicated each other – the language network was not implicated in processing programming code. The replication substantially mitigates the concerns shared by us and the reviewers about sample size and control tasks. The fact that different control tasks and sample sizes did not change the overall pattern of results, in our view, is affirmation of the robustness of the findings, and the value that both submissions presented together can offer the literature.

In sum, there were concerns that both submissions were exploratory in nature, lacking a strong theoretical focus, and relied on functional localizers on novel tasks. However, these concerns were mitigated by the following strengths. Both tasks ask a clear and interesting question. The results replicate each other despite task differences. In this way, the two papers strengthen each other. Specifically, the major concerns for each paper individually are ameliorated when considering them as a whole.

In your revisions, please address the concerns of the reviewers, including, specifically, the limits of interpretation of your results with regard to control task choice, the discussion of relevant literature mentioned by the reviewers, and most crucially, please contextualize your results with regard to the other submission's results.

Reviewer #1:

This manuscript is clearly written and the methods appear to be rigorous, although the number of subjects (15) is a bit low; however, this does not appear to critically limit interpretation of the results. I appreciated the focused inclusion on expert coders to make a clear comparison to language. I also thought that the inclusion of multiple domains for comparison (logic, math, executive function, and language) was quite informative. The laterality covariance between code and language was also quite interesting. I do have some concerns with the literature review and discussion of present and previous results.

1) My main concern with this paper is that it does not clearly review previous fMRI studies on code processing. How do the present results compare with previous studies? E.g. Castelhano et al., 2019; Floyd et al., 2017; Huang et al., 2019; Krueger et al., 2020; Siegmund et al., 2017, 2014;) It seems like the localization/lateralization obtained in the present study is largely similar to these previous studies (e.g. Siegmund et al., 2017). If so, this should discussed: a convergence across multiple methods/authors is useful to know. Any discrepancies are also useful to know. The authors suggest that "Moreover, no prior study has directly compared the neural basis of code to other cognitive domains." However, Krueger et al., (2020) and Huang et al., (2019) appear to have done this.

2) The authors should point out and discuss the difficulty of understanding the psychological and neural structure of coding in absence of a clear theory of coding, as is the case for language (e.g. Chomsky, 1965; Levelt, 1989; Lewis and Vasishth, 2005). On this point, I appreciate the reference to Fitch et al., (2005) regarding recursion in coding, but I think it would be most helpful to have a clear example of recursion in python code. However, the authors at least focus their results on neural underpinnings without attempting to make strong claims about cognitive underpinnings.

3) The authors’ report overlap between code comprehension and language in the posterior MTG and IFG. They note that these activations were somewhat inconsistent; yet they did observe this significant overlap. However, the paper discusses the results as if this overlap did not occur, e.g. "We find that the perisylvian fronto-temporal network that is selectively responsive to language, relative to math, does not overlap with the neural network involved in code comprehension." This is not accurate, as there indeed was overlap. It is important to point out that among language-related regions, these two regions are the most strongly associated with abstract syntax (Friederici, 2017; Hagoort, 2005; Tyler and Marslen-Wilson, 2008; Pallier et al., 2011; Bornkessel-Schlesewsky and Schlesewsky, 2013; Matchin and Hickok, 2019), which very well could be a point of shared resources among code and language (as discussed in Fitch, 2005).

Reviewer #2:

The goal of this fMRI study was to determine which brain systems support coding, by way of extent of overlap of univariate maps with localizer tasks for language, logic, math, and executive functions. The basic conclusion is one we could have anticipated: coding engages a widespread frontoparietal network, with stronger involvement of the left hemisphere. It overlaps with all of the other tasks, but most with the map for logic. This doesn't seem too surprising, but the authors argue convincingly that others wouldn't have predicted that.

It's unfortunate that there are differences in task difficulty among the tasks, in particular, that the logic task was the most difficult of all (both in terms of accuracy and response times), since that happens to be the one that had the largest number of overlapping voxels with the coding task. We can't know whether coding and language task voxels would have overlapped more if the language task had been more difficult.

It seems a shame to present data only from highly experienced coders (11+ years of experience); I can imagine that the investigators are planning to write up another study examining effects of expertise, in comparison with less experienced coders. This seems like an initial paper that's laying the groundwork for a more groundbreaking one.

Reviewer #3:

This fMRI study examines an interesting question, namely how computer code – as a "cognitive/cultural invention" – is processed by the human brain. However, I have a number of concerns with regard to how this question was examined in terms of experimental design, including the choice of control condition (fake code) and the way in which localiser tasks were utilised. In addition, the sample size is very small (n=15) and there appear to be large inter-individual differences in coding performance (in spite of the recruitment of expert programmers). In summary, while promising in its aims, the study's conclusions are weakened by these considerations related to its execution.

1) The control condition

The experiment contrasted real Python code with fake code in the form of "incomprehensible scrambled Python functions". Real and fake code also differed in regard to the task performed (code comprehension versus memory) and were distinguished via colour coding. There is a lot to unpack here in regard to how processing might differ between the two different conditions. For example, the real-code blocks required code comprehension as well as computational problem solving (which does not necessarily require the use of code), while the control task requires neither. As a result of the colour coding, it also appears likely that participants will have approached the fake code blocks with a completely different processing strategy than the real-code blocks. These are just a few obvious differences between the conditions but there are likely many more given how different they are. This, in my view, makes it difficult to interpret the basic contrast between real and fake code.

2) Use of localiser tasks

A similar concern as for point 1 holds in regard to the localiser tasks that were used in order to examine anatomical overlap (or lack thereof) between code comprehension and language, maths, logical problem solving and multiple-demand executive control, respectively. I am generally somewhat sceptical in regard to the use of functional localisers in view of the assumptions that necessarily enter into the definition of a localiser task. This concern is exacerbated by the way in which localisers were employed in the present study. Firstly, in addition to the definition of the localiser task itself, this study used localiser contrasts to define networks of interest. For example, the contrast language localiser > maths localiser served to define the "language network". Thus, assumptions about the nature of the localiser itself are compounded with those regarding the nature of the contrast. Secondly, particularly with regard to language, the localiser task was very high level, i.e. requiring participants to judge whether an active and a passive sentence had the same meaning (with both statements remaining on the screen at the same time). While of course requiring language processing, this task is arguably also a problem solving task of sorts. It is certainly more complex than a typical task designed to probe fast and automatic aspects of natural language processing.

In addition, given that reading is also a cultural invention, is it really fair to say that coding is being compared to the "language network" here rather than to the "reading network" (in view of the visual presentation of the language task)? The possible implications of this for the interpretation of the data should be considered.

More generally, while an anatomical overlap between networks active during code comprehension and networks recruited during other cognitive tasks may shed some initial light on how the brain processes code, it doesn't support any particularly strong conclusions about the neural mechanisms of code processing in my view. While code comprehension may overlap anatomically with regions involved in executive control and logic, this doesn't mean that the same neuronal populations are recruited in each task nor that the processing mechanisms are comparable between tasks.

3) Sample size and individual differences

At n=15, the sample size of this study is quite small, even for a neuroimaging study. This again limits the conclusions that can be drawn from the study results.

Moreover, the results of the behavioural pre-test – which was commendably included – suggest that participants differed considerably with regard to their Python expertise. For the more difficult exercise in this pre-test, the mean accuracy score was 64.6% with a range from 37.5% to 93.75%. These substantial differences in proficiency weren't taken into account in the analysis of the fMRI data and, indeed, it appears difficult to meaningfully do so in view of the sample size.

---

## [Author Response]

Reviewer #1:This manuscript is clearly written and the methods appear to be rigorous, although the number of subjects (15) is a bit low; however, this does not appear to critically limit interpretation of the results. I appreciated the focused inclusion on expert coders to make a clear comparison to language. I also thought that the inclusion of multiple domains for comparison (logic, math, executive function, and language) was quite informative. The laterality covariance between code and language was also quite interesting. I do have some concerns with the literature review and discussion of present and previous results.1) My main concern with this paper is that it does not clearly review previous fMRI studies on code processing. How do the present results compare with previous studies? E.g. Castelhano et al., 2019; Floyd et al., 2017; Huang et al., 2019; Krueger et al., 2020; Siegmund et al., 2017, 2014;) It seems like the localization/lateralization obtained in the present study is largely similar to these previous studies (e.g. Siegmund et al., 2017). If so, this should discussed: a convergence across multiple methods/authors is useful to know. Any discrepancies are also useful to know. The authors suggest that "Moreover, no prior study has directly compared the neural basis of code to other cognitive domains." However, Krueger et al., (2020) and Huang et al., (2019) appear to have done this.

In response to the reviewer’s suggestion we have added a paragraph in the Introduction which reviews prior literature on code processing. We also added a section in the Discussion integrating the current results with the co-published article by Ivanova and colleagues. Overall, the current results are consistent with the findings of Ivanova et al., in finding fronto-parietal responses to code. The convergence and differences among the current article and that of Ivanova is discussed in detail, with regard to lateralization and localization.

Overall, it seems that the current results are convergent with prior work. However, with regard to studies prior to the Ivanova paper, it is difficult to say conclusively because of the differences between the current study and prior research. The handful of prior studies that have examined the neural bases of coding have used more complex tasks (e.g. writing code, debugging code, evaluating proposed code edits), rather than code comprehension. Control conditions in prior studies are different from the current study (e.g. mental rotation) and often complex (e.g. prose editing). In several cases the contrasts that would enable comparing to the current study are not reported or the methods and analyses are not described in detail, perhaps partly because all prior findings are reported in computer science proceedings journals with different reporting practices. Below we summarize the studies noted by the reviewer in particular.

Huang et al., (2019) subtracted neural activation associated with the mental rotation of three-dimensional objects from neural activation associated with manipulation of two types of data structures (sequence and binary search tree). For sequences, but not for binary trees, greater fronto-parietal activity was observed the code tasks than the mental rotation task. This study did not directly examine code comprehension per se and did not localize functions, such as language or working memory in the same group of participants.

Krueger et al., (2020) compared code writing to prose writing, either in a fill-in-the-blank format, or in the format of free writing in response to a prompt. Analyses compared code and prose directly using three contrasts: all code > all prose, fill-in-the-blank code > fill-in-the-blank prose, and free code > free prose. As we now note in the Introduction, this study observed fronto-parietal activity for code relative to prose and is thus generally consistent with the current findings.

Note that neither Huang et al., (2019) or Krueger et al., (2020) localized specific previously documented neural networks and compared code comprehension to these networks. Therefore, they reported the difference in neural basis between code and other cognitive tasks, but not their shared neural resources.

Siegmund et al., (2017) compared comprehension of different types of Java code. In particular, they compared comprehension of code with meaningful and meaningless variable and function names and compared code comprehension of both types to detection of syntactic errors within code. These comparisons are not specifically designed to identify networks involved in code comprehension per se. A collection of regions was more responsive during code comprehension than syntactic error detection, some of these regions may overlap with the current fronto-parietal system. However, no coordinate table is reported, and the authors say that some of the areas may overlap with language networks. Direct comparisons are difficult. We now specifically point to the Siegmund et al., 2017 and 2014 papers in the Introduction.

2) The authors should point out and discuss the difficulty of understanding the psychological and neural structure of coding in absence of a clear theory of coding, as is the case for language (e.g. Chomsky, 1965; Levelt, 1989; Lewis and Vasishth, 2005). On this point, I appreciate the reference to Fitch et al., (2005) regarding recursion in coding, but I think it would be most helpful to have a clear example of recursion in python code. However, the authors at least focus their results on neural underpinnings without attempting to make strong claims about cognitive underpinnings.

We agree with the reviewer’s point that a theory of coding is needed to properly understand its neurocognitive basis. We added text to the Discussion section pointing this out.

In response to the reviewer’s suggestion, we also added examples of recursion in computer code in the Introduction. Specifically, we mentioned the following examples:

IF conditionals embedded within IF conditionals:

if (condition_1):

if (condition_2):

print(“Both conditions are True.”)

else:

print(“Condition_1 is True, condition_2 is False.”)

else:

print(“Condition_1 is False. Condition_2 not evaluated.”)

A function calling itself in its definition:

def factorial(N):

return N*factorial(N-1) if (N>1) else 1

3) The authors’ report overlap between code comprehension and language in the posterior MTG and IFG. They note that these activations were somewhat inconsistent; yet they did observe this significant overlap. However, the paper discusses the results as if this overlap did not occur, e.g. "We find that the perisylvian fronto-temporal network that is selectively responsive to language, relative to math, does not overlap with the neural network involved in code comprehension." This is not accurate, as there indeed was overlap. It is important to point out that among language-related regions, these two regions are the most strongly associated with abstract syntax (Friederici, 2017; Hagoort, 2005; Tyler and Marslen-Wilson, 2008; Pallier et al., 2011; Bornkessel-Schlesewsky and Schlesewsky, 2013; Matchin and Hickok, 2019), which very well could be a point of shared resources among code and language (as discussed in Fitch, 2005).

In response to the reviewer’s suggestion, we changed the wording in the Discussion to more narrowly state that there was low and variable overlap between code comprehension and language, rather than no overlap. It is worth noting that half of the participants showed overlap that was no different from chance. Nevertheless, we now also point out that the overlapping regions have been implicated in syntactic processing, as well as in semantics and that we cannot rule out the possibility that the language network plays some role in code comprehension

Reviewer #2:The goal of this fMRI study was to determine which brain systems support coding, by way of extent of overlap of univariate maps with localizer tasks for language, logic, math, and executive functions. The basic conclusion is one we could have anticipated: coding engages a widespread frontoparietal network, with stronger involvement of the left hemisphere. It overlaps with all of the other tasks, but most with the map for logic. This doesn't seem too surprising, but the authors argue convincingly that others wouldn't have predicted that.

We agree with the reviewer that overlap between the activation for code comprehension and logical reasoning makes a lot of sense but is not represented in the literature.

It's unfortunate that there are differences in task difficulty among the tasks, in particular, that the logic task was the most difficult of all (both in terms of accuracy and response times), since that happens to be the one that had the largest number of overlapping voxels with the coding task. We can't know whether coding and language task voxels would have overlapped more if the language task had been more difficult.

We agree with the reviewer that the existence of a difference in difficulty among the localizer tasks is less than ideal. However, in the current study it is also intrinsic to the design. Language is a putatively evolutionarily ancient function that is acquired by children without explicit instruction early in life. By contrast, formal logical reasoning is an explicit skill that requires teaching and is not acquired by most educated adults. Therefore, these cognitive domains differ in difficulty by design. We could make the language task more difficult artificially, for example by constructing complex grammatical sentences. However, such sentences are not representative of the type of processing we typically do with language. More importantly, previous evidence suggests that when language gets difficult in this way, non-language specific networks get recruited for processing. Despite the fact that the language task is easier, large swaths of cortex are responsive more to language than math, suggesting that responses in this particular network are not related to task difficulty per se.

Finally, because we worried about the same issue, in our analyses we control for the number of overall active vertices in each contrast. Overlap is calculated as the ratio between “the number of overlapping vertices” and “the square root of the product of the numbers of vertices in both the code contrast and each localizer contrast”, thus normalizing for the number of vertices involved in the contrasts. Additionally, we plot the degree of overlap for each task with itself separately and show the degree of overlap with code relative to this measure.

It seems a shame to present data only from highly experienced coders (11+ years of experience); I can imagine that the investigators are planning to write up another study examining effects of expertise, in comparison with less experienced coders. This seems like an initial paper that's laying the groundwork for a more ground-breaking one.

We appreciate the reviewer’s insight into the future direction of this research project. The current study is indeed an initial paper which serves as the groundwork for further studies. Among the many possible follow-up studies, examining the effects of expertise is definitely one of the most exciting.

Reviewer #3:This fMRI study examines an interesting question, namely how computer code – as a "cognitive/cultural invention" – is processed by the human brain. However, I have a number of concerns with regard to how this question was examined in terms of experimental design, including the choice of control condition (fake code) and the way in which localiser tasks were utilised. In addition, the sample size is very small (n=15) and there appear to be large inter-individual differences in coding performance (in spite of the recruitment of expert programmers). In summary, while promising in its aims, the study's conclusions are weakened by these considerations related to its execution.1) The control conditionThe experiment contrasted real Python code with fake code in the form of "incomprehensible scrambled Python functions". Real and fake code also differed in regard to the task performed (code comprehension versus memory) and were distinguished via colour coding. There is a lot to unpack here in regard to how processing might differ between the two different conditions. For example, the real-code blocks required code comprehension as well as computational problem solving (which does not necessarily require the use of code), while the control task requires neither. As a result of the colour coding, it also appears likely that participants will have approached the fake code blocks with a completely different processing strategy than the real-code blocks. These are just a few obvious differences between the conditions but there are likely many more given how different they are. This, in my view, makes it difficult to interpret the basic contrast between real and fake code.

We agree with the reviewer that the real-code condition and the fake code condition were different in various ways. Since this is one of the first studies to examine code comprehension, the experiment was designed to maximally capture potentially interesting neural structures involved in code comprehension by having a relatively low-level control condition that removes basic reading processes and working memory demands. For this reason, we also presented real-codes and fake codes in different colors such that the participants are not trying to figure out whether the stimulus is or is not real-code, contaminating the neural response to fake codes with attempted code comprehension. We now explicitly point out these goals in the Materials and methods section.

Because the contrast between code and fake code is large, it was also important to show that patterns of activity in the code-sensitive network can distinguish IF and FOR code functions. In future work, it will be important to dig more deeply into the question of what and how the identified network is processing. We now specifically point this out in subsection “Code overlaps with logic”. In particular, as the reviewer correctly points out, the current paper does not determine whether the code-sensitive network identified is involved in computer coding algorithms (what the reviewer calls computational problem solving) or code relevant syntax. In the coding literature this is sometimes referred to as the semantics vs. syntax of code (e.g. citation). Both of these types of processes are part and parcel of code comprehension, although only some of them are unique to code. The results of the Ivanova study speak to this question to some degree. We now discuss this issue in the Discussion section.

2) Use of localiser tasksA similar concern as for point 1 holds in regard to the localiser tasks that were used in order to examine anatomical overlap (or lack thereof) between code comprehension and language, maths, logical problem solving and multiple-demand executive control, respectively. I am generally somewhat sceptical in regard to the use of functional localisers in view of the assumptions that necessarily enter into the definition of a localiser task. This concern is exacerbated by the way in which localisers were employed in the present study. Firstly, in addition to the definition of the localiser task itself, this study used localiser contrasts to define networks of interest. For example, the contrast language localiser > maths localiser served to define the "language network". Thus, assumptions about the nature of the localiser itself are compounded with those regarding the nature of the contrast. Secondly, particularly with regard to language, the localiser task was very high level, i.e. requiring participants to judge whether an active and a passive sentence had the same meaning (with both statements remaining on the screen at the same time). While of course requiring language processing, this task is arguably also a problem solving task of sorts. It is certainly more complex than a typical task designed to probe fast and automatic aspects of natural language processing.

We appreciate the reviewer’s point that localizer approaches have limitations. In our view, for the present purposes this was the best approach available, despite these limitations. Nevertheless, the approach has consequences for the inferences that can be made based on the present results. We now talk about this issue in the Discussion section:

“Notably, in the current study neural responses associated with language, math and logic were localized partly relative to each other (e.g. logic > language). This approach focuses the analyses on networks that are uniquely involved in one domain vs. another. The degree to which areas shared across language, logic, and math are implicated in code comprehension remains to be addressed in future work.”

With regard to the reviewer’s specific comment about the language task, we agree that there is a problem solving component. However, this component is present also in the other control tasks (i.e. logic and math) and is matched as closely as possible across the tasks. Furthermore, the language-related responses observed in the current study are consistent with previous studies using other tasks and contrasts, including comparing passive sentence comprehension to a non-word control task (Fedorenko et al., 2011, 2016). Perhaps even more directly relevant, Ivanova and colleagues used a different language localizer task but found results consistent with the current study. This is now discussed in detail in the Discussion section.

In addition, given that reading is also a cultural invention, is it really fair to say that coding is being compared to the "language network" here rather than to the "reading network" (in view of the visual presentation of the language task)? The possible implications of this for the interpretation of the data should be considered.

We appreciate the reviewer’s point regarding the fact that the current language task used reading, rather than spoken language comprehension. Notably, we have previously done the exact same language task using spoken language stimuli and observed the same neural network responsive to spoken language (Kanjlia et al., 2016). Since the control tasks are also written in English characters, the reading-specific activation is likely similar in the language and control conditions.

More generally, while an anatomical overlap between networks active during code comprehension and networks recruited during other cognitive tasks may shed some initial light on how the brain processes code, it doesn't support any particularly strong conclusions about the neural mechanisms of code processing in my view. While code comprehension may overlap anatomically with regions involved in executive control and logic, this doesn't mean that the same neuronal populations are recruited in each task nor that the processing mechanisms are comparable between tasks.

We agree with the reviewer’s point that the current results do not support precise conclusions about the cognitive or neural processes involved in code comprehension, beyond comparison to the tasks described in the study (see also response to reviewer 1, comment 2). In future work we plan to use the findings of the current study as a necessary step towards more detailed understanding of the neural and cognitive basis of code. In particular, we agree that future work will need to examine closely distribution of responses to code within fronto-parietal systems. We now point out in the Discussion that future work will need to examine whether there is sub-specialization within the fronto-parietal network for code as opposed to other functions.

3) Sample size and individual differencesAt n=15, the sample size of this study is quite small, even for a neuroimaging study. This again limits the conclusions that can be drawn from the study results.

We updated the manuscript to point out the sample size limitation. (Discussion section)

Moreover, the results of the behavioural pre-test – which was commendably included – suggest that participants differed considerably with regard to their Python expertise. For the more difficult exercise in this pre-test, the mean accuracy score was 64.6% with a range from 37.5% to 93.75%. These substantial differences in proficiency weren't taken into account in the analysis of the fMRI data and, indeed, it appears difficult to meaningfully do so in view of the sample size.

As the reviewer points out, there is variability among participants’ expertise. Since coding, unlike language, is a cultural invention acquired in adulthood, such variability is to be expected. Absence of variability would likely indicate insensitivity of the testing measurement, rather than absence of variability. We agree with the reviewer, that given the small sample size, the current study is not suited to comprehensively testing effects of expertise level. We did test whether expertise predicted lateralization patterns and it did not. However, this conclusion is limited by the sample size. An investigation into the individual difference in programming expertise, and its influence on the neural response to code comprehension will be one of our future research topics.